# Pathomorphological Manifestations and the Course of the Cervical Cancer Disease Determined by Variations in the *TLR4* Gene

**DOI:** 10.3390/diagnostics13121999

**Published:** 2023-06-07

**Authors:** Eglė Žilienė, Arturas Inčiūra, Rasa Ugenskienė, Elona Juozaitytė

**Affiliations:** 1Institute of Oncology, Lithuanian University of Health Sciences, LT-50161 Kaunas, Lithuania; arturas.inciura@lsmu.lt (A.I.); rasa.ugenskiene@lsmu.lt (R.U.); elona.juozaityte@lsmu.lt (E.J.); 2Department of Genetics and Molecular Medicine, Lithuanian University of Health Sciences, LT-50161 Kaunas, Lithuania

**Keywords:** cervical cancer, *TLR4*, polymorphisms, genotype, metastases, survival

## Abstract

Cervical cancer (CC) is often associated with human papillomavirus (HPV). Chronic inflammation has been described as one of the triggers of cancer. The immune system fights diseases, including cancer. The genetic polymorphism of pathogen recognition receptors potentially influences the infectious process, development, and disease progression. Many candidate genes SNPs have been contradictory demonstrated to be associated with cervical cancer by association studies, GWAS. *TLR4* gene activation can promote antitumor immunity. It can also result in immunosuppression and tumor growth. Our study aimed to investigate eight selected polymorphisms of the *TLR4* gene (rs10759932, rs1927906, rs11536898, rs11536865, rs10983755, rs4986790, rs4986791, rs11536897) and to determine the impact of polymorphisms in genotypes and alleles on the pathomorphological characteristics and progression in a group of 172 cervical cancer subjects with stage I–IV. Genotyping was performed by RT-PCR assay. We detected that the CA genotype and A allele of rs11536898 were significantly more frequent in patients with metastases (*p* = 0.026; *p* = 0.008). The multivariate logistic regression analysis confirmed this link to be significant. The effect of rs10759932 and rs11536898 on progression-free survival (PFS) and overall survival (OS) has been identified as important. In univariate and multivariate Cox analyses, AA genotype of rs11536898 was a negative prognostic factor for PFS (*p* = 0.024; *p* = 0.057, respectively) and OS (*p* = 0.008; *p* = 0.042, respectively). Rs11536898 C allele predisposed for longer PFS (univariate and multivariate: *p* = 0.025; *p* = 0.048, respectively) and for better OS (univariate and multivariate: *p* = 0.010; *p* = 0.043). The worse prognostic factor of rs10759932 in a univariate and multivariate Cox analysis for survival was CC genotype: shorter PFS (*p* = 0.032) and increased risk of death (*p* = 0.048; *p* = 0.015, respectively). The T allele of rs10759932 increased longer PFS (univariate and multivariate: *p* = 0.048; *p* = 0.019, respectively) and longer OS (univariate and multivariate: *p* = 0.037; *p* = 0.009, respectively). Our study suggests that SNPs rs10759932 and rs11536898 may have the potential to be markers contributing to the assessment of the cervical cancer prognosis. Further studies, preferably with larger groups of different ethnic backgrounds, are needed to confirm the results of the current study.

## 1. Introduction

In today’s oncology, the genetic features of the host that determine the pathophysiology of cancer and the course of the disease are intensively studied. The genetic influence on cancer is multifunctional. The risk of cancer is increased by additional factors that activate the immune system and cause inflammation. Inflammatory mediators can contribute to neoplasia by inducing mutations, adaptive responses, resistance to apoptosis, and environmental changes such as stimulation of angiogenesis [1,2,3].

Scientific studies suggest that membrane-associated innate Toll-like receptors (TLRs) as pattern recognition receptors (PRRs) play a main role in activating the immune response associated with autoimmune diseases, inflammation, and tumor-associated diseases. The human TLRs family consists of 10 members (TLR1–TLR10). They are expressed in human immune cells and many tumors. Each of their expressions elicits a different response. These are transmembrane proteins that can recognize pathogen-associated molecular patterns (PAMPs) or host damage-associated molecular patterns (DAMPs) to activate innate and adaptive immune responses by triggering activation of NF-κB, AP1, CREB, c/EBP, and IRF transcription factors. TLRs mediate changes in the expression of chemokines and pro-inflammatory cytokines and activate the response of cytotoxic lymphocytes, thereby eliminating pathogens and host debris [4,5,6,7]. The signaling pathway of TLRs begins in the cytoplasmic TIR domain, which contains adaptors such as MyD88, TIRAP, and TRIF that modulate TLR signaling pathways, helping to recognize antigenic molecules (lipopolysaccharides, nucleic acids). This activates the protein complex, such as NF-kB, IRFs, MAP kinases, via the MyD88-dependent way on the recruitment of members of the IRAK family, TRIF-dependent way or MyD88-independent pathway. It regulates the production of cytokines, chemokines, type I interferon, thus eliminating antigens. Negative regulation of the signal path helps protect the host from inflammatory damage [8,9,10,11]. Studies have shown that TLRs can produce the desired antitumor effects by inducing apoptosis, autophagy, and necrosis in tumor cells [12,13,14]. TLR expression correlates with cancer prognosis [15,16]. Activation of TLRs becomes a target for cancer immunotherapy [17,18,19,20,21,22].

*TLR4* gene, which consists of three exons and is localized on chromosome 9q33.1, is one of the most studied TLRs. Mutations in the *TLR4* gene have been shown to induce resistance of pathogens to lipopolysaccharides in mice [23]. *TLR4* mutations are associated with endotoxin hyporesponsiveness in humans [24]. The TLR4 receptor is likely to be associated with several diseases because of the range of ligands (both pathogen-related and endogenous) identified as agonists of *TLR4* [25]. *TLR4* is linked to a range of diseases with potential treatments targeting the *TLR4* pathway [26,27,28,29,30,31,32,33,34,35,36,37,38,39,40,41,42,43,44,45,46,47,48,49,50,51,52,53,54,55,56,57,58,59,60,61,62,63,64,65,66,67]. TLR4 activation can not only cause antitumor immunity but also, conversely, promote immunosuppression and influence tumor growth [68]. Changes in *TLR4* gene expression are involved in carcinogenesis. Activated *TLR4* increases inflammatory cytokines and cell proliferation, migration, invasion, and survival. Overexpression of *TLR4* in malignant cells promotes tumor growth and metastases [69]. High expression of *TLR4* is likely associated with the poor survival outcome of patients with solid cancers [70]. Cervical cancer, which starts in the cervix, is a widespread health problem and one of the most common oncological diseases. According to the World Health Organization, it is the fourth most commonly diagnosed cancer in women. It also ranks fourth in the world for cancer-related deaths among women. There were an estimated 604,000 new cases of cervical cancer and 342,000 deaths from the disease worldwide in 2020. *TLR4* promotes HPV-positive cervical tumor growth and facilitates the formation of a local immunosuppressive microenvironment. These conditions may lead to CC development [71]. *TLR4* expression was reported in accordance with histopathological grade in human papillomavirus (HPV)-infected cervical cells: the level was higher in invasive cervical cancers (ICC) than in cervical intraepithelial neoplasia (CIN) and low in normal cervical tissues. Moreover, higher *TLR4* expression in HPV-positive cervical cancer cell lines SiHa and HeLa, compared with the HPV-negative cell line C33A, was observed, indicating a role for HPV infection in *TLR4* regulation [72]. A link between increased *TLR4* expression and the severity of cervical lesions was found and was closely associated with FIGO stage, lymph node metastases, and tumor size in CC. In the advanced stages of FIGO, larger tumor sizes, and higher *TLR4* expression levels were observed [73]. Various factors influencing the development of CC have been identified. Cervical tumorigenesis is often initiated by high-risk HPV [74,75]. 

Single nucleotide polymorphism (SNP) is likely to affect cancer susceptibility. The influence of polymorphisms of the *TLR4* gene on various cancerous diseases was investigated [76,77,78,79]. To comprehensively analyze the impact of germinal polymorphisms on the course of the disease, the main components influencing the spread of cancer are investigated. A review of the global literature focused on the influence of *TLR4* gene polymorphisms and expression on the course of various cancers (tumor proliferation, differentiation, metastases, prognosis, and patient survival). An association between *TLR4* polymorphisms and a risk of hypersensitivity to HPV16/18 infection in women and increased risk of cervicitis, the precancerous lesion, has been identified [80,81,82,83,84]. However, there are very few studies on the impact of *TLR4* gene polymorphisms on the pathomorphological features or course of cervical cancer.

The *TLR4* gene was chosen for the analyses as it encodes the protein that interferes with HPV, leading to dysregulation of the local immune microenvironment and tumorigenesis. With limited reports on the role of *TLR4* polymorphisms in cervical cancer, we performed a study to investigate eight selected polymorphisms of the *TLR4* gene (rs10759932, rs1927906, rs11536898, rs11536865, rs10983755, rs4986790, rs4986791, rs11536897), whose links to other cancers from other research have been published. We analyzed the distribution of polymorphisms in genotypes and alleles in a group of patients with cervical cancer, and we determined the correlations between SNPs and tumor pathomorphological parameters and the course of the disease.

## 2. Materials and Methods

### 2.1. Study Subjects

The study of interest in cervical cancer was approved by the Kaunas Regional Biomedical Research Ethics Committee (No. BE-2-10 and P1-BE-2-10/2014). One hundred seventy-two adult patients treated at the Hospital of Lithuanian University of Health Sciences Kauno Clinics with stage I–IV of cervical cancer, who agreed to participate in the retrospective study, were enrolled consecutively. Patient exclusion criteria were other malignancies and incomplete medical documentation. A written informed consent was obtained from all the participants. Subjects were recruited from October 2014 to August 2020. The follow-up period was until November 2020. All patients were treated according to standard protocols. The vast majority underwent chemoradiation therapy (69.2%), while others underwent surgery followed by radiotherapy or systemic treatment.

### 2.2. Methods

Clinical data on participants, tumor morphological characteristics, and the course of the disease were collected from the medical records. The cervical cancer diagnosis was made by a gynecologist performing a gynecological and radiological examination, and based on the histopathology of cervical biopsies. All carcinoma cases were staged following the recommendations of the International Federation of Gynecology and Obstetrics (FIGO). The tumor grading system was based on architectural and cytologic (nuclear) criteria. The age at the time of diagnosis, tumor size (T), lymph node involvement (N), the spread of metastases (M), stage, differentiation degree (G), presence of disease progression, and death of patients were considered as clinicopathological features in this analysis.

### 2.3. SNP Selection

Genotype data were identified using online databases—The International HapMap Project (http://www.HapMap.org, accessed on 1 October 2020) and the 1000 Genomes Project (http://www.1000genomes.org, accessed on 1 October 2020). The selection criteria of *TLR4* SNPs were as follows: SNPs were detected in other populations, related to the outcomes of different diseases reported in other studies, SNPs have not been widely analyzed before among patients with CC, a minor allele frequency (MAF) of SNP was ≥5% in the European population, and SNPs might be a functional site mapped.

### 2.4. DNA Extraction and Genotyping

DNA was extracted from leucocytes of peripheral venous blood samples collected in vacuum tubes with ethylenediaminetetraacetate (EDTA) and stored in a laboratory biobank at −20 °C. DNA genotyping was performed at the Institute of Oncology (Lithuanian University of Health Sciences). Genomic DNA extraction was performed using a genomic DNA purification Kit (Thermo Fisher Scientific Baltics, Vilnius, Lithuania). SNPs in the *TLR4* gene were determined using the TaqMan^®^ probe SNP Genotyping Assay (Thermo Fisher Scientific, Baltics, Vilnius, Lithuania). Molecular genetic studies were performed using the real-time polymerase chain reaction (RT-PCR) method. PCR was used to amplify a particular segment of DNA based on the protocol. The candidate SNPs, location, region, MAF in the European population, and the primers used for RT-PCR are listed in Table 1.

## 3. Results

### 3.1. Tumour Characteristics and SNP Frequencies

In our study, 90.1% of the participants were of Lithuanian nationality, while the remaining nationalities were Polish, Estonian, Ukrainian, and German. The majority of patients were aged ≥50 years old (71.5%) and had the squamous-type histopathology variant of cervical cancer (92.3%). Other rare histopathology variants, including adenocarcinoma, adenosquamous cell carcinoma, mucinous adenocarcinoma, accounted for 8.7% of the subjects. The vast majority of tumors (65.7%) were moderately differentiated (G2). The most commonly diagnosed stage was stage IIB cancer (32.0%). Slightly less than half of the subjects (44.8%) had pathological regional lymph nodes. Nine patients had pathologic paraaortic lymph nodes, which were categorized as stage IIIC2 cancer. Distant metastases were diagnosed in ten patients. The detailed distribution of clinicopathological features can be found in the Table 2.

We performed Hardy–Weinberg equilibrium testing for each SNP. Three of the SNPs did not follow Hardy–Weinberg equilibrium (HWE). In the cases of SNP rs11536865, all the cases had the GG genotype. For SNP rs10983755, the distribution of genotype was as follows: GG—92.4%, GA—7.6%, AA—0%. For SNP rs11536897, the distribution of genotype was as follows: GG—96.0%, AG—4.0%, AA—0%. Others SNPs distribution of genotypes was as follows: Rs10759932 TT—70.4%, TC—27.3%, CC—4%; rs1927906 TT—77.3%, TC—22.1%, CC—0.6%; rs11536898 CC—75.0%, CA—22.1%, AA—2.9%; Rs4986790 AA—86.0%, AG—13.4%, GG—0.6%; Rs4986791 CC—85.5%, CT—13.9%, TT—0.6%; The total count and frequencies of TLR4 genotypes and alleles are presented in Table 3.

### 3.2. Association Analysis

In our study, we analyzed the potential associations between SNPs and tumor clinicopathological features. However, no statistically significant correlations between rs10759932, rs1927906, rs11536865, rs10983755, rs4986790, rs4986791, rs11536897, and tumor size, nodes status, metastases, tumor cells differentiation, stage were found performing logistic regression. All the analyzed polymorphisms were not related to the patients’ age at the time of diagnosis (*p* > 0.05). Nevertheless, we detected a significant link between SNP rs11536898 and metastases (M). Carrying the A allele statistically significantly increased the chance of having metastases (OR = 5.068, 95% CI: 1.357–18.918, *p* = 0.008). This finding was partially confirmed by the genotype model, as patients with the CA genotype had a 4.735 higher risk for distal metastases than patients with the CC genotype (95% CI: 1.204–18.626, *p* = 0.026). This may be due to the fact that only five patients with the AA genotype were determined in our study, which may have affected *p*-value in this comparison. All the results are presented in Table 4 and Table 5. Furthermore, the multivariate logistic regression analysis confirmed the significant link between rs11536898 and metastases. In the multivariate analysis (Model No.1), the CA genotype significantly increased the risk to having metastases (OR = 4.609, 95% CI: 1.166–18.212, *p* = 0.029), and the A allele increased the risk for metastases (OR = 5.044, 95% CI: 1.346–18.899, *p* = 0.016) when adjusting for the age group at the diagnosis. The relationship remains statistically significant when adjusting for the age at the diagnosis and tumor differentiation (G) (Model No.2): the CA genotype significantly increased the risk of having metastases (OR = 4.419, 95% CI: 1.111–17.576, *p* = 0.035). Additionally, the A allele increased the risk for metastases (OR = 4.884, 95% CI: 1.297–18.392, *p* = 0.019). We did not include tumor size and nodules in multivariate models because we observed complete data separation (Table 6).

### 3.3. Survival Analysis

In our study group, disease progression was observed in 30.2% of patients during the follow-up period. The localization of progression were as follows: local progression was found in 13.36% of all subjects, positive lymph nodes (N+) were detected in 3.66%, local progression and positive nodes were found in 4.08%, and in other cases (9.3%), progression included local progression, positive lymph nodes, and distant metastases. There were 40 instances of death (23.3%) during the follow-up period. In all cases, the cause of death was cancer progression.

The effect of the SNPs on survival (PFS and OS) was analyzed in the genotype and allelic models. The PFS ranged from 1 to 201 months (median 13). The OS also went from 1 to 201 months (median 16.5). No significant link between rs1927906, rs11536865, rs10983755, rs4986790, rs4986791, rs11536897 genotypes or alleles and survival was detected. However, the effect of two SNPs (rs10759932 and rs11536898) on PFS and OS has been identified as important. The Kaplan—Meier method showed a link between rs10759932 CC genotype and OS (*p* = 0.049 for Log Rank, *p* = 0.018 for Breslow, and *p* = 0.028 for Tarone–Ware). Cox’s regression analysis demonstrated the influence of the CC genotype on shorter PFS and OS compared to patients with the TT genotype (OR = 2.918, 95% CI: 0.894–9.530, *p* = 0.049; OR = 3.340, 95% CI: 1.006–11.095, *p* = 0.048, respectively). Our multivariate Cox’s regression model included tumor T, N, G, and the age of patients. In the adjusted analysis, the CC genotype increased the risk of progression by almost four times compared to the TT genotype (OR = 3.674, 95% CI: 1.115–12.108, *p* = 0.032) and increased the risk of faster mortality (OR = 4.608, 95% CI: 1.344–15.801, *p* = 0.015). The T allele was significant for PFS (Log Rank *p* = 0.049, Breslow *p* = 0.042, Tarone–Ware *p* = 0.042) and OS (Log Rank *p* = 0.031, Breslow *p* = 0.018, Tarone–Ware *p* = 0.023) in the allelic model. Carrying the T allele increased the possibility of longer PFS (OR = 0.331, 95% CI: 0.103–1.067, *p* = 0.048) and longer OS (OR = 0.284, 95% CI: 0.087–0.928, *p* = 0.037). The holders of T allele had an increased chance of longer PFS (OR = 0.244, 95% CI: 0.075–0.795, *p* = 0.019) and a decreased risk of shorter OS (OR = 0.200, 95% CI: 0.059–0.674, *p* = 0.009) when the adjustment for tumor T, N, G and the age of patients (Table 7 and Table 8).

The rs11536898 AA genotype subgroup, compared to the CC genotype, was also significantly associated with PFS (Log Rank, *p* = 0.014, Breslow *p* = 0.001, Tarone–Ware *p* = 0.003) and OS (Log Rank, *p* = 0.003, Breslow *p* < 0.001, Tarone–Ware *p* < 0.001). The rs11536898 AA genotype compared to patients with the CC genotype decreased the likelihood for longer PFS (OR = 3.926, 95% CI: 1.201–12.837, *p* = 0.024) and shortened OS (OR = 5.057, 95% CI: 1.522–16.802, *p* = 0.008). In the multivariate Cox’s regression analysis, the AA genotype remained a factor that shortened OS (OR = 3.735, 95% CI: 1.051–13.278, *p* = 0.042) and showed a borderline effect on PFS (OR = 3.306, 95% CI: 0.967–11.299, *p* = 0.057), when the adjustment for tumor T, N, G and the age of patients. The rs11536898 C allele was significantly associated with PFS (Log Rank, *p* = 0.015, Breslow *p* = 0.003, Tarone–Ware *p* = 0.005) and OS (Log Rank, *p* = 0.004, Breslow *p* < 0.001, Tarone–Ware *p* = 0.001). No significant effect of the CA genotype on PFS was determined. This is in line with the allelic model, which demonstrated that the carriers of the C allele were less likely to have shorter PFS compared to non-carriers. The rs11536898 C allele predisposed to longer PFS (OR = 0.261, 95% CI: 0.081–0.844, *p* = 0.025) and longer OS (OR = 0.212, 95% CI: 0.065–0.691, *p* = 0.010). When adjusting for tumor T, N, G, and age of patients, the tendency remains statistically significant for PFS (OR = 0.291, 95% CI: 0.086–0.987, *p* = 0.048) and for OS (OR = 0.274, 95% CI: 0.078–0.959, *p* = 0.043) (Table 7 and Table 8).

Kaplan—Meier analysis was performed to generate survival curves for genotypes and alleles for both PFS and OS (Figure 1).

## 4. Discussion

The active investigation of the correlation between *TLR4* SNPs and CC is intriguing. The current study is the first to investigate analyzed SNPs in assessing the clinicopathological features as well as the course of CC. It establishes the relationship between SNPs in *TLR4* and CC, suggesting their potential as biomarkers that could be used for prognosticating the development of the disease. In the future, SNP detection in *TLR4* may be used to stratify patients, predict clinical manifestations of CC, assess risks of progression or relapse, and evaluate treatment efficacy. The study has many advantages: the dataset contains genetic data, tumor phenotype data, and survival information. However, there are limitations in our study. We cannot compare our results with others because we did not find any studies analyzing associations between these polymorphisms and clinicopathological characteristics of CC. In addition, our results may have been affected by the limited sample size. We hope to expand the study group in the future. Another weakness of this study is the lack of a control group to assess CC risk. Genotyping errors are expected to be minor. Thus, the resulting biases will likely to be small.

Two of the SNPs (rs10759932 and rs11536898) were significant in our analysis. Rs10759932 is located in the promoter region of the *TLR4* gene and may regulate the *TLR4* expression level by influencing the binding affinity of transcription factors [85]. We found that the rare homozygous rs10759932 CC genotype causes shorter PFS and OS. The allelic model did not contradict the survival results. The T allele showed the link to better survival, although the effect of the C allele on worse survival prognosis was not statistically significant. However, previously published studies provide contradictory data for rs10759932 correlation with cancers. Some researchers’ findings could contribute to our study’s results, indicating that the CC genotype is an indicator of a worse outcome. T. Tongtawee et al. investigated 400 patients with gastric lesions, including chronic gastritis, gastric atrophy, internal metaplasia, and gastric cancer. They found that the rs10759932 CC homozygous genotype significantly increased the risk of premalignant and malignant gastric lesion development [86]. The Cleveland case-control study in Caucasians and African Americans supported the influence of rs10759932 on prostate cancer risk. Men carrying the CC genotype for rs10759932 had a statistically significant increased risk of prostate cancer (*p* = 0.006) compared to men carrying the TT genotype [87]. On the other hand, other research provides opposite results. The study conducted in the Shandong Province of Northern China demonstrates that the rs10759932 polymorphism was associated with susceptibility to gastric cancer (GC) in both genotype and allelic frequency. However, genotype CC was the protective factor for GC. They believe that the genetic variant of *TLR4* rs10759932 might play an essential protective role in the development of GC [88]. Huang et al. found that the rs10759932 TC heterozygote and combined genotypes TC/CC were associated with a significantly reduced risk of gastric cancer in the high-risk population [85]. Similar results were obtained from a Japanese study where the rs10759932 TC/CC genotypes decreased the risk of gastric cancer. However, this did not reach statistical significance (*p* = 0.059) [89]. Several studies have not shown any correlations between rs10759932 polymorphisms and cancer. These include studies on breast cancer in the Saudi population [90], the risk of noncardia gastric cancer in Goyang [91], and the risk of colorectal cancer in Brazil [92]. A large nested case-control study of prostate cancer in the Physicians’ Health Study (1982–2004), including 1267 controls and 1286 random prostate cancer cases, showed that genetic variation across this polymorphism is not strongly associated with prostate cancer risk or mortality [93]. The dissociation of research results possibly be due to sample size limitations, different ethnic groups, and the multicausal backgrounds promoting cancer development, including genetic factors, race, environment, and lifestyle.

In our study, rs11536898 was also associated with CC clinical outcomes. The rare AA genotype causes shorter PFS and OS compared to the CC genotype. The C allele was inversely associated with shorter PFS and OS. In addition, the AC genotype and A allele were associated with an increased risk of metastases. On the contrary, a significant association was revealed with the risk of prostate cancer. The Health Professionals Follow-up Study (HPFS) found that men under the age of age 65 carrying two copies of the minor alleles of rs11536898 had a statistically significantly lower risk of prostate cancer compared to noncarriers (CC and CA versus AA: OR 0.59; 95% CI 0.41–0.86) [94]. However, many of the SNPs in this study were in high linkage disequilibrium with one another. The Physicians‘ Health Study found no statistically significant association between this SNP and the overall prevalence of prostate cancer. In addition, there were no significant associations between the SNP and cases of advanced/fatal or severe cancer, and there was also no evidence of associations between *TLR4* SNPs and prostate cancer-specific mortality or bone metastases [93]. Observational results from another population-based case-control study showed that rs11536898 was associated with colon cancer, where the AA vs. CA/CC genotype decreased colon cancer risk (OR 0.50, 95% CI 0.29, 0.87) [95]. However, other studies (the Washington County Cancer Registry, the Maryland Cancer Registry, Sweden, the Physicians’ Health Study) found no significant interfaces between rs11536898 and cancer [93,96,97]. Although the data are unclear, we believe that the A allele may be associated with a worse prognosis.

Unfortunately, in our study, we did not find statistically significant associations with pathomorphological features or outcomes of cervical cancer for SNPs rs4986790 and rs4986791, which have been widely studied worldwide and are potentially associated with other cancers, influencing the risk or prognosis. Rs4986790 is a common polymorphism that causes an amino acid exchange (aspartate to glycine). In a study involving 122 Tunisian women with cervical cancer compared with 260 healthy control, the *TLR4* polymorphism Asp299Gly (rs4986790) was found to be associated with a higher risk of cervical cancer. The common homozygote Asp/Asp genotype and the Asp allele were associated with a higher risk of developing cervical cancer (OR 4.95, CI: 1.97–13.22) and (OR 5.17, CI: 2.11–13.50), respectively [98]. Another Tunisian case-control study with 130 cervical cancer patients and 260 controls showed that the rs4986790 dominant genotype Asp/Asp was significantly more frequent among cervical cancer cases with early stage (I + II) and advanced stage (III + IV) than controls. The major allele Asp was a risk factor for the I + II stage tumors [99]. Opposite results were reported in an Indian cervical cancer study involving 110 untreated cervical cancer patients and 141 healthy controls, where the minor allele G of rs4986790 was associated with an increased risk of cervical cancer, although a genotypic association was not found [83]. Pandey et al., in a study of North Indian women, did not observe an association between rs4986790 and rs4986791 with cervical cancer risk at the genotype, allele, and haplotype level. However, this study with 150 cervical cancer patients and 150 healthy female controls provided data showing that the Thr399Ile (rs4986791) polymorphism Thr/Ile genotype was significantly associated (*p* = 0.044) with Stage II cervical cancer and conferred a 2.51 fold risk of developing cervical cancer at an early stage [100]. A Chinese Han population study with 1262 participants, including 420 cervical cancer patients and 842 controls, did not find any significant association of rs4986791 with cervical cancer risk [101]. In the evaluation of other female-related cancers, the allele and genotype frequencies for the polymorphism rs4986790 were compared between 191 endometrial cancer cases and 291 controls in a study at the Hunter Centre, Australia, but no associations were observed for endometrial cancer risk [102]. The TLR4 Asp299Gly and Thr399Ile alleles were not detected in the 105 ovarian cancer patients in a study conducted in northern China. These results indicate that the *TLR4* 299Gly and 399Ile alleles were exhibited at a lower frequency in northern Chinese ovarian cancer patients compared to other studies [103]. Among the 70 women with ovarian cancer enrolled in the Poland study, the heterozygous variant and the recessive G allele of rs4986790 were more frequently found than in the 130 healthy individuals, indicating an increased risk of OC for its carriers. No difference in the distribution of rs4986791 between the cases and controls was observed [104]. A study conducted at the “Hippocratic” General Hospital of Athens, Greece, which included 261 breast cancer patients and 480 healthy individuals, showed that Gly carriers of rs4986790 (Asp/Gly & Gly/Gly genotype) and the Gly allele were more common among the breast cancer cases (*p* = 0.0031 and *p* = 0.0061, respectively) [105]. It was found to have a significant association with breast cancer malignancy in the ER-patient groups for the rs4986790 in the Saudi Arabian population. In the ER-group, the AA genotype presented a significantly higher frequency in the patients compared to the controls. Similarly, the genotype AG was considerably less frequent in the cases compared to the controls [90].

*TLR4* polymorphisms rs4986790 and rs4986791 may be associated with a significantly increased gastric cancer risk. Two publications by Juliana Garcia de Oliveira mention the significant influence of these SNPs on gastric cancer in the Brazilian population [106,107]. However, a study by Garza-Gonzalez et al. reported no correlation between *TLR4* polymorphisms and gastric cancer in the Mexican population [108]. A study by Trejo-de la et al. in a Mexican population found that the D299G (rs4986790) polymorphism was significantly associated with duodenal ulcer and showed a trend for association with gastric cancer [109]. In an Italian population case-control study by Santini et al. the Thr399Ile polymorphism was linked to increased susceptibility to gastric cancer [110]. Caucasian population-based case-control study data suggest that the *TLR4* + 896A > G polymorphism is a risk factor for non-cardia gastric carcinoma and its precursors [111]. The frequency of risk alleles of rs4986790 and rs4986791 in a nested case-control gastric cancer study in the European Prospective Study Cancer Group was low, and they could only estimate the association in the codominant model, which did not show significance [112]. In an Ethnic Kashmiri Population, no significance was observed in the appearance of gastric cancer, but odds Ratio analysis showed that carriers of the Asp299Gly G allele were significantly associated with the tumors in the distal part of the stomach. In contrast, carriers of the Thr399Ile T allele were associated with well-differentiated gastric adenocarcinoma [113]. The overall results from a meta-analysis of gastric cancer risk suggest that *TLR4* polymorphisms (+896A/G and +1196C/T) may be associated with a significantly increased risk of gastric cancer in Caucasians [114]. However, no SNPs were found at the sites Asp299Gly and Thr399Ile to be associated with susceptibility to GC in the Shangdong Province of Northern China [88]. 

Several studies have analyzed the association of rs4986790 and rs4986791 with colorectal cancer (CRC). However, studies conducted in Brazilian, Irish, Danish, and Iranian populations did not find a significant effect on CRC risk [92,115,116,117]. A meta-analysis supported the association of TLR4 genetic polymorphisms with an increased risk of CRC among Asians but not among Caucasians and Africans stratified by ethnic group [118]. An Egyptian study revealed that rs4986790 G allele carriers were more frequent in the CRC group compared to controls, and the T allele of rs4986791 was associated with an increased risk for CRC. Additionally, the rs4986791 CT/TT genotype was observed to be significantly linked to CRC. The G allele of rs4986790 predisposed to CRC progression, including high cancer stage IV, high grade III, positive lymph nodes (N2), and metastases [119]. The study conducted on Russian individuals with various solid tumors suggested that the A/G genotype for the rs4986790 SNP correlated with an 80% increased colorectal cancer. Rs4986790 polymorphisms were more evident in patients with rectal cancer separately [120].

Large case-control studies, meta-analysis have shown no significant association between rs4986790, rs4986791, and prostate cancer risk or clinical features [92,97,121,122].

Only a few studies have analyzed the associations of these SNPs with other cancers. A Turkish case-control study on lung cancer (NSCLC, SCLC) indicated that rs4986790 was not associated with lung cancer. However, the 3.857-fold risk was evaluated for the rs4986791 CT genotype compared to CC in lung cancer (*p* = 0.041) [123]. A study on patients with head and neck squamous cell carcinomas from Germany reported the results of the investigated SNPs rs4986790 and rs4986791. The Asp299Gly genotype, compared to Asp299Asp, was associated with poorer DFS (*p* = 0.04) and worse OS (*p* = 0.04). Patients with the rs4986790 wild-type genotype (*TLR4* Asp299Asp vs. *TLR4* Asp299Gly) had significantly longer DFS with adjuvant systemic treatment (*p* = 0.004). For another SNP rs4986791, a similar pattern was observed in DFS [124]. In a case-control study from the Indian population, rs4986791 was associated with a significantly elevated risk of gallbladder cancer [125]. Between melanoma cases and controls in Germany, patients carrying the minor allele for the rs4986790 polymorphism were associated with prolonged overall survival (*p* = 0.01) and survival following metastases (*p* = 0.02) [126]. SNPs rs4986790 and rs1927906 were genotyped in a study of Saudi Arabian patients with acute lymphoblastic leukemia (ALL) and healthy controls. Only the AG showed a significant association with a protective effect against ALL (*p* = 0.002) [127].

Several meta-analyses have attempted to generalize these data. In a meta-analysis by Zhu L et al. based on 34 publications, *TLR4* rs4986790 and rs4986791 were found to increase overall cancer risk. The effect of rs4986790 on cancer risk was more evident in female-specific cancers and digestive cancers, especially for gastric cancer. The risks effect of rs4986791 was also prominent in gastric cancer. However, no significant association was observed between rs4986790 and prostate cancer risk. The association between rs4986791 and cancer risk was significant in both South Asians and East Asians, but not in Caucasians [128]. Ding et al. represented the results of a meta-analysis based on 55 publications. They found that Rs4986790 was not strongly associated with cancer risk. Meanwhile, the rs4986791 polymorphism has always been associated with reduced cancer risk in the general population. Moreover, they found that Caucasian female-specific cancers were significantly associated with rs4986790 polymorphisms in subgroup analysis by cancer type and race, while Asian digestive cancers were significantly influenced by the rs4986791 polymorphism [77]. 

Accumulated evidence has implicated *TLR4* polymorphism in modulating the risk and development of various types of cancers. However, we still need to replicate those findings. In our study, the results of these SNPs did not serve as indicators of possible disease progression biomarker.

We found no association of rs10983755, rs11536897, rs11536865, rs1927906 SNPs with clinical features and outcomes of CC. The rs10983755 polymorphism affects the risk of gastric carcinogenesis and can provide some protection against H. pylori infection [91,129]. However, the results showed no significant association with H. pylori infection, and there was no significant association of any examined genotype with the overall survival of GC in another Chinese population study. Patients with lymph node metastases undergoing postoperative chemotherapy and carrying the rs10983755 AA genotype had an HR of 0.328, compared to those carrying the GG + AG genotype [130]. The *TLR4* rs11536897 (−3084), rs1927906 (3189), and rs11536865 (−729G/C) polymorphisms are rare SNPs, and their functions remain unclear. In our study, all cases of rs11536865 had GG genotypes. There is a suggestion of an interaction between polymorphisms within *TLR4* and the HCV status [131]. The −729GC polymorphism was associated with an increased risk of bladder cancer. Moreover, the −729GC genotype significantly affected lower TLR4 mRNA and protein levels, suggesting that the polymorphism may lead to dysregulation of TLR4 expression, interfering with TLR4 promoter activity [132]. In a Korean men‘s case-control prostate cancer study, all 300 cases revealed the GG genotype at rs11536897 [133]. No link between SNPs rs11536897, rs1927906, and prostate cancer was found in the pooled Sweden case-control study and meta-analysis by Weng et al. [97,121]. 

The rs1927906 heterozygous CT was associated with a decreased cancer risk in Saudi Arabian patients with an ALL [127].

## 5. Conclusions

Our study suggests that SNPs rs10759932 and Rs11536898 may have the potential to be markers contributing to the assessment of cervical cancer survival prognosis. The SNP rs11536898 prompts us to consider its impact on cancer metastases and further research in this area. However, due to the limited sample size, a larger group of patients with cervical cancer is required to confirm the obtained results. Our results provide insight for future studies on cervical cancer and other infection-related cancer types, which can evaluate these polymorphisms to determine their functionality. Although evidence is accumulating for the importance of genetic variation in the etiology and development of cervical cancer, research investigating the role of immune-related gene variants in cervical cancer is still in its early stages. Further studies, preferably with larger groups of individuals from different ethnic backgrounds, are needed to confirm the results of the current study. Therefore, identifying the variants responsible for maintaining the tumor immune response may provide more specific targets to combat cervical cancer development and disease progression. In the future, SNP detection in TLR4 may be used to predict the clinical manifestations, risk, and prognosis of CC.

## Figures and Tables

**Figure 1 diagnostics-13-01999-f001:**
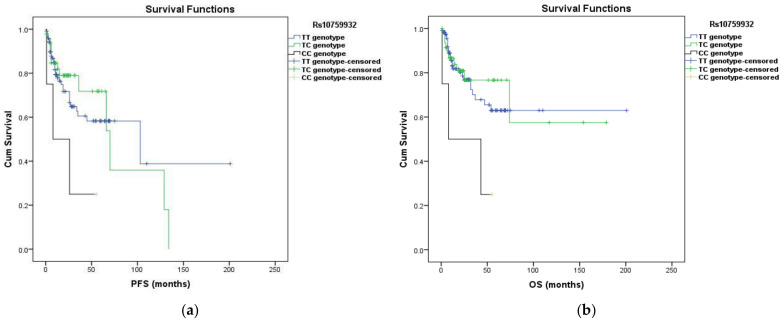
Kaplan—Meier survival curves for PFS and OS in patients with cervical cancer according rs10759932 and rs11536898 polymorphisms (n = 172). Kaplan—Meier survival curves for rs10759932 and rs11536898 polymorphisms in the genotype and allelic models demonstrating PFS and OS differences. The *X*-axis displays the number of months from cervical cancer, confirming the event date (PFS or OS), and the *Y*-axis indicates the survival probability. (**a**,**b**) Rs10759932 CC genotype increased the risk for shorter PFS and OS compared to patients with the TT genotype (95% CI: 0.894–9.530, *p* = 0.049; 95% CI: 1.006–11.095, *p* = 0.048, respectively); (**c**,**d**) Carrying the T allele increased the possibility of longer PFS (95% CI: 0.103–1.067, *p* = 0.048) and longer OS (95% CI: 0.087–0.928, *p* = 0.037); (**e**,**f**) Rs11536898 AA genotype compared to patients with the CC genotype shortened PFS (95% CI: 1.201–12.837, *p* = 0.024) and OS (95% CI: 1.522–16.802, *p* = 0.008); (**g**,**h**) Rs11536898 C allele predisposed to longer PFS (95% CI: 0.081–0.844, *p* = 0.025) and to longer OS (95% CI: 0.065–0.691, *p* = 0.010).

**Table 1 diagnostics-13-01999-t001:** SNPs genomic region, MAF.

SNP	Genomic Position in chr9	Region/Location	MAF/Highest Population MAF
rs10983755 G > A	117702392	Promoter, 5′-UTR, intergenic variant −2081	0.07/0.31 (A)
rs10759932 T > C	117702866	Promoter, 5′-UTR, −1607	0.18/0.35 (C)
rs11536865 G > C	117703745	Promoter, 5′-UTR, regulatory region variant −729	0.04/0.24 (C)
rs4986790 A > G Asp299Gly	117713024	Exon, 3′-UTR, missense variant 896	0.06/0.14 (G)s
rs4986791 C > T Thr399Ile	117713324	Exon, 3′-UTR, missense variant 1196	0.04/0.17 (T)
rs11536897 G > A	117717732	3′-UTR 3084	0.04/0.11 (A)
rs1927906 T > C	117717837	3′-UTR 3189	0.21/0.49 (C)
rs11536898 C > A	117717932	3′-UTR 3284	0.13/0.27 (A)

**Table 2 diagnostics-13-01999-t002:** General clinicopathological characteristics and frequencies of 172 study participants.

Variables	Subgroups	Frequencies (Count/%)
Age (years) (mean ± SD: 55.4 ± 13.5)	≥50	123/71.5%
	<50	49/28.5%
Histology	Squamous	157/92.3%
	Non-squamous	15/8.7%
Tumor size (T)	T1A	1/0.6%
	T1B	25/14.6%
	T2A	4/2.3%
	T2B	80/46.5%
	T3A	13/7.6%
	T3B	38/22.1%
	T3C	4/2.3%
	T4A	4/2.3%
	T4B	3/1.7%
Pathological regional lymph nodes status	N0	95/55.2%
	N1	77/44.8%
Distant metastases	M0	162/94.2%
	M1	10/5.8%
Stage	IA	1/0.6%
	IB	15/8.7%
	IIA	5/2.9%
	IIB	12/7.0%
	IIIA	9/5.2%
	IIIB	12/7.0%
	IIIC1	53/31.0%
	IIIC2	9/5.2%
	IVA	3/1.7%
	IVB	10/5.8%
Grade	1	13/7.6%
	2	113/65.7%
	3	46/26.7%

**Table 3 diagnostics-13-01999-t003:** The distribution of *TLR4* genotypes and alleles.

SNP	Genotypes Frequencies			Alleles Frequencies	
**Rs10759932 T > C**	TT	121	0.704	T	0.841
	TC	47	0.273	C	0.159
	CC	4	0.023		
**Rs1927906 T > C**	TT	133	0.773	T	0.883
	TC	38	0.221	C	0.117
	CC	1	0.006		
**Rs11536898 C > A**	CC	129	0.750	C	0.861
	CA	38	0.221	A	0.139
	AA	5	0.029		
**Rs11536865 G > C**	GG	172	1.000	G	1.000
	GC	0	0		
	CC	0	0		
**Rs10983755 G > A**	GG	159	0.924	G	0.962
	GA	13	0.076	A	0.038
	AA	0	0		
**Rs4986790 A > G**	AA	148	0.860	A	0.927
	AG	23	0.134	G	0.073
	GG	1	0.006		
**Rs4986791 C > T**	CC	147	0.854	C	0.924
	CT	24	0.140	T	0.076
	TT	1	0.006		
**Rs11536897 G > A**	GG	165	0.959	G	0.979
	GA	7	0.041	A	0.021
	AA	0	0		

**Table 4 diagnostics-13-01999-t004:** Univariate logistic regression analysis between SNP’s and tumor characteristics.

		Possitive T3–T4 Versus T1–2			Possitive N1 Versus N0			Possitive M1 Versus M0			Possitive G3 Versus G1 + G2		
SNP	Genotype, alleles	OR	95% CI	*p*-Value	OR	95% CI	*p*-Value	OR	95% CI	*p*-Value	OR	95% CI	*p*-Value
**Rs10759932**	TC vs. TT	1.276	0.639–2.551	0.490	1.272	0.647–2.499	0.486	2.158	0.554–8.413	0.268	1.110	0.520–2.370	0.787
	CC vs. TT	1.881	0.256–13.834	0.535	1.327	0.181–9.734	0.781	7.733	0.678–88.176	0.099	2.903	0.392–21.495	0.297
	T allele + vs. T -	0.570	0.078–4.150	0.574	0.806	0.111–5.861	0.831	0.170	0.016–1.800	0.097	0.355	0.049–2.596	0.290
	C allele + vs. C -	1.317	0.6673–2.577	0.421	1.276	0.662–2.460	0.467	2.522	0.697–9.122	0.147	1.210	0.584–2.504	0.608
**Rs1927906**	TC vs. TT	1.033	0.489–2.183	0.932	0.668	0.318–1.403	0.567	1.543	0.379–6.278	0.545	1.919	0.887–4.154	0.098
	CC vs. TT	0.000	0.000	1.000	0.000	0.000	1.000	0.000	0.000	1.000	0.000	0.000	1.000
	T allele + vs. T -	2.758	2.261–3.364	0.187	2.250	1.903–2.660	0.265	1.062	1.023–1.103	0.803	3.800	2.957–4.883	0.097
	C allele + vs. C -	1.107	0.530–2.310	0.787	0.716	0.345–1.484	0.368	1.500	0.369–6.097	0.325	2.056	0.962–4.398	0.060
**Rs11536898**	CA vs. CC	1.405	0.671–2.944	0.368	1.543	0.746–3.191	0.242	4.735	1.204–18.626	**0.026**	1.576	0.722–3.439	0.253
	AA vs. CC	2.898	0.469–17.989	0.253	2.083	0.337–12.898	0.430	7.812	0.704–86.710	0.094	0.758	0.082–7.030	0.807
	C allele + vs. C -	0.374	0.061–2.300	0.271	0.530	0.086–3.258	0.487	0.228	0.023–2.254	0.169	1.475	0.161–13.555	0.730
	A allele + vs. A -	1.529	0.757–3.090	0.235	1.597	0.798–3.197	0.184	5.068	1.357–18.918	**0.008**	1.463	0.689–3.106	0.320
**Rs10983755**	AG vs. GG	1.088	0.340–3.482	0.887	1.483	0.477–4.613	0.496	1.389	0.162–11.900	0.764	2.550	0.809–8.035	0.110
	AA vs. GG	*	*	*	*	*	*	*	*	*	*	*	*
	G allele + vs. G -	0.000	0.000	1.000	0.000	0.000	1.000	0.000	0.000	1.000	0.000	0.000	1.000
	A allele + vs. A -	1.088	0.340–3.482	0.887	1.483	0.477–4.613	0.494	1.389	0.162–11.900	0.763	2.550	0.809–8.035	0.110
**Rs4986790**	AG vs. AA	1.152	0.467–2.841	0.758	0.501	0.195–1.289	0.152	1.667	0.331–8.388	0.536	1.600	0.628–4.077	0.325
	GG vs. AA	0.000	0.000	1.000	0.000	0.000	1.000	0.000	0.000	1.000	0.000	0.000	1.000
	A allele + vs. A -	2.758	2.261–3.364	0.187	2.250	1.903–2.660	0.265	1.062	1.023–1.103	0.803	3.800	2.957–4.883	0.267
	G allele + vs. G -	1.280	0.532–3.082	0.581	0.572	0.231–1.419	0.225	1.591	0.317–7.986	0.570	1.800	0.727–4.455	0.199
**Rs4986791**	TC vs. CC	1.305	0.542–3.143	0.553	0.581	0.234–1.441	0.241	1.580	0.315–7.929	0.580	1.486	0.588–3.756	0.402
	TT vs. CC	0.000	0.000	1.000	0.000	0.000	1.000	0.000	0.000	1.000	0.000	0.000	1.000
	C allele + vs. C -	2.758	2.261–3.364	0.187	2.250	1.903–2.660	0.265	1.062	1.023–1.103	0.803	3.800	2.957–4.883	0.267
	T allele + vs. T -	1.435	0.608–3.389	0.408	0.653	0.271–1.573	0.340	1.511	0.302–7.567	0.613	1.672	0.682–4.103	0.258
**Rs11536897**	AG vs. GG	0.682	0.128–3.623	0.653	0.922	0.200–4.251	0.917	0.000	0.000	1.000	1.100	0.206–5.877	0.911
	AA vs. GG	*	*	*	*	*	*	*	*	*	*	*	*
	G allele + vs. G -	0.000	0.000	1.000	0.000	0.000	1.000	0.000	0.000	1.000	0.000	0.000	1.000
	A allele + vs. A -	0.682	0.128–3.623	0.651	0.922	0.200–4.251	0.917	0.939	0.904–0.977	0.502	1.100	0.206–5.877	0.911

T3–T4, T1–2—tumor size, N1—pathological regional lymph nodes, N0—no pathological regional lymph nodes, M1—distant metastases, M0—no distant metastases, G1 + G2, G3—tumor grade. * OR could not be estimated because of zero value within a cell.

**Table 5 diagnostics-13-01999-t005:** Univariate logistic regression analysis: the odds ratio for associations between SNPs and the age of patients, stage groups, and expected prognosis of the disease.

		Possitive Stage III–IV Versus Stage I–II			Possitive Worse Prognosis: T3–T4 + G3 Versus T1–T2 + G1–G2			Age (Years): ≤50 vs. >50		
SNP	Genotype, Alleles	OR	95% CI	*p*-Value	OR	95% CI	*p*-Value	OR	95% CI	*p*-Value
**Rs10759932**	TC vs. TT	1.088	0.551–2.148	0.808	1.143	0.396–3.298	0.805	1.575	0.793–3.134	0.195
	CC vs. TT	0.806	0.110–5.911	0.832	4.000	0.237–67.473	0.336	5.854	0.590–58.052	0.131
	T allele + vs. T -	1.270	0.175–9.232	0.813	0.259	0.016–4.304	0.346	0.195	0.020–1.915	0.120
	C allele + vs. C -	1.062	0.549–2.055	0.857	1.273	0.463–3.500	0.640	1.734	0.891–3.377	0.104
**Rs1927906**	TC vs. TT	0.655	0.317–1.350	0.251	2.159	0.785–5.940	0.136	0.896	0.393–1.178	0.642
	CC vs. TT	0.000	0.000	1.000	0.000	0.000	1.000	0.000	0.000	1.000
	T allele + vs. T -	1.800	1.574–2.058	0.372	0.000	0.000	1.000	2.672	2.201–3.243	0.378
	C allele + vs. C -	0.691	0.338–1.414	0.310	2.429	0.907–6.506	0.078	0.900	0.429–1.891	0.852
**Rs11536898**	CA vs. CC	.1.123	0.541–2.334	0.755	1.725	0.579–5.140	0.328	1.227	0.588–2.563	0.586
	AA vs. CC	1.225	0.198–7.582	0.827	2.156	0.184–25.271	0.541	0.422	0.046–3.885	0.446
	C allele + vs. C -	0.838	0.136–5.146	0.848	0.524	0.045–6.045	0.604	2.485	0.272–22.734	0.651
	A allele + vs. A -	1.135	0.564	2.281	1.776	0.631–4.997	0.277	1.103	0.544–2.240	0.856
**Rs10983755**	AG vs. GG	1.291	0.405–4.119	0.666	2.430	0.535–11.030	0.250	1.453	0.466–4.528	0.520
	AA vs. GG	*	*	*	*	*	*	*	*	*
	G allele + vs. G -	0.000	0.000	1.000	0.000	0.000	1.000	0.000	0.000	1.000
	A allele + vs. A -	1.291	0.405–4.119	0.666	2.430	0.535–11.030	0.250	1.453	0.366–4.528	0.520
**Rs4986790**	AG vs. AA	0.570	0.235–1.383	0.214	2.005	0.616–6.533	0.248	0.876	0.349–2.199	0.778
	GG vs. AA	0.000	0.000	1.000	0.000	0.000	1.000	0.000	0.000	1.000
	A allele + vs. A -	1.800	1.574–2.058	0.372	0.000	0.000	1.000	0.000	0.000	1.000
	G allele + vs. G -	0.627	0.264–1.492	0.289	2.406	0.781–7.416	0.126	0.986	0.405–2.402	0.975
**Rs4986791**	TC vs. CC	0.635	0.267–1.510	0.304	2.005	0.616–6.533	0.248	0.812	0.327–2.022	0.655
	TT vs. CC	0.000	0.000	1.000	0.000	0.000	1.000	0.000	0.000	1.000
	C allele + vs. C -	1.800	1.574–2.058	0.372	0.000	0.000	1.000	0.000	0.000	1.000
	T allele + vs. T -	0.692	0.296–1.620	0.395	2.406	0.781–7.416	0.126	0.914	0.378–2.208	0.842
**Rs11536897**	AG vs. GG	0.581	0.126–2.677	0.486	0.000	0.000	1.000	0.000	0.000	1.000
	AA vs. GG	*	*	*	*	*	*	*	*	*
	G allele + vs. G -	0.000	0.000	1.000	0.000	0.000	1.000	0.000	0.000	1.000
	A allele + vs. A -	0.581	0.126–2.677	0.481	*	*	*	*	*	*

* OR could not be estimated because of zero value within a cell.

**Table 6 diagnostics-13-01999-t006:** Multivariate logistic regression analyses for metastases adjusted for Rs11536898 genotype, age at the diagnosis and tumor differentiation (G).

			Model No.1			Model No.2		
Dependent	SNP	Covariates	Odds	95%CI	*p*	Odds	95%CI	*p*
Positive M	Rs11536898	CA vs. CC	4.609	1.166–18.212	**0.029**	4.419	1.111–17.576	**0.035**
		AA vs. CC	9.452	0.803–111.217	0.074	9.871	0.827–117.76	0.070
		Age group	0.977	0.928–1.028	0.370	0.977	0.928–1.029	0.376
		Possitive G3 vs. G1+G2				1.729	0.445–6.716	0.429
			**Model No.1**			**Model No.2**		
**Dependent**	**SNP**	**Covariates**	**Odds**	**95%CI**	** *p* **	**Odds**	**95%CI**	** *p* **
Positive M	Rs11536898	A allele + vs. A -	5.044	1.346–18.899	**0.016**	4.884	1.297–18.392	**0.019**
		Age group	0.979	0.931–1.030	0.415	0.980	0.932–1.030	0.426
		Possitive G3 vs. G1+G2				1.670	0.433–6.439	0.456

**Table 7 diagnostics-13-01999-t007:** Cox’s univariate model for PFS and OS.

			Progression-Free Survivol			Overall Survival	
SNP	Genotype/Allele	HR	95% CI	*p*-Value	HR	95% CI	*p*-Value
**Rs10759932**	TC vs. TT	0.884	0.472–1.653	0.699	0.818	0.382–1.752	0.606
	CC vs. TT	2.918	0.894–9.530	**0.049**	3.340	1.006–11.095	**0.048**
	T allele + vs. T -	0.331	0.103–1.067	**0.048**	0.284	0.087–0.928	**0.037**
	C allele + vs. C -	1.012	0.564–1.816	0.967	1.012	0.509–2.010	0.973
**Rs1927906**	TC vs. TT	0.975	0.498–1.910	0.975	0.695	0.306–1.576	0.383
	CC vs. TT	2.584	0.352–18.949	0.350	3.081	0.417–22.761	0.383
	T allele + vs. T -	0.385	0.053–2.807	0.346	0.301	0.041–2.216	0.239
	C allele + vs. C -	1.028	0.537–1.971	0.933	0.770	0.354–1.673	0.509
**Rs11536898**	CA vs. CC	1.103	0.586–2.073	0.762	1.294	0.636–2.633	0.476
	AA vs. CC	3.926	1.201–12.837	**0.024**	5.057	1.522–16.802	**0.008**
	C allele + vs. C -	0.261	0.081–0.844	**0.025**	0.212	0.065–0.691	**0.010**
	A allele + vs. A -	1.274	0.707–2.295	0.420	1.545	0.803–2.971	0.193
**Rs10983755**	AG vs. GG	0.508	0.123–2.097	0.349	0.341	0.043–2.290	0.253
	AA vs. GG	*	*	*	*	*	*
	G allele + vs. G -	0.000	0.000	1.000	0.000	0.000	1.000
	A allele + vs. A -	0.508	0.123–2.097	0.349	0.341	0.043–2.290	0.253
**Rs4986790**	AG vs. AA	1.482	0.716–3.069	0.290	1.062	0.444–2.542	0.892
	GG vs. AA	2.767	0.378–20.275	0.316	3.346	0.453–24.696	0.236
	A allele + vs. A -	0.385	0.053–2.807	0.346	0.301	0.041–2.216	0.239
	G allele + vs. G -	1.554	0.774–3.123	0.215	1.178	0.520–2.669	0.695
**Rs4986791**	TC vs. CC	1.426	0.689–2.952	0.339	1.029	0.430–2.461	0.950
	TT vs. CC	2.752	0.376–20.165	0.319	3.331	0.451–24.582	0.238
	C allele + vs. C -	1.499	0.746–3.010	0.256	0.301	0.041–2.216	0.239
	T allele + vs. T -	0.385	0.053–2.807	0.346	1.142	0.504–2.587	0.750
**Rs11536897**	AG vs. GG	0.425	0.058–3.084	0.397	1.314	0.316–5.454	0.707
	AA vs. GG	*	*	*	*	*	*
	G allele + vs. G -	0.000	0.000	1.000	0.000	0.000	1.000
	A allele + vs. A -	0.425	0.058–3.084	0.397	1.314	0.316–5.454	0.707

* OR could not be estimated because of zero value within a cell.

**Table 8 diagnostics-13-01999-t008:** Cox’s multivariate model for PFS and OS. The adjusted ratio for associations between SNPs rs10759932 and rs11536898 and age at the time of diagnosis, tumor characteristics.

	Variables	Progression-Free Survivol			Overall Survival		
		HR	95% CI	*p*-Value	HR	95% CI	*p*-Value
**Rs10759932**	TC vs. TT	0.658	0.338–1.280	0.217	0.747	0.351–1.590	0.449
	CC vs. TT	3.674	1.115–12.108	**0.032**	4.608	1.344–15.801	**0.015**
	Age at diagnosis	0.993	0.971–1.016	0.566	1.017	0.991–1.043	0.199
	T3-T4 vs. T1-T2	5.540	2.870–10.694	<0.001	8.178	3.489–19.167	<0.001
	N1 vs. N0	1.340	0.709–2.534	0.368	1.775	0.854–3.689	0.124
	G3 vs. G1-2	0.913	0.490–1.704	0.776	0.773	0.384–1.556	0.471
**Rs10759932**	T allele + vs. T -	0.244	0.075–0.795	**0.019**	0.200	0.059–0.674	**0.009**
	Age at diagnosis	0.996	0.973–1.018	0.697	1.018	0.993–1.044	0.163
	T3-T4 vs. T1-T2	5.298	2.750–10.206	<0.001	8.045	3.430–18.871	<0.001
	N1 vs. N0	1.291	0.684–2.439	0.431	1.735	0.835–3.604	0.140
	G3 vs. G1-2	0.962	0.520–1.779	0.902	0.797	0.399–1.593	0.521
**Rs11536898**	CA vs. CC	0.858	0.440–1.675	0.654	1.090	0.522–2.277	0.819
	AA vs. CC	3.306	0.967–11.299	**0.057**	3.735	1.051–13.278	**0.042**
	Age at diagnosis	0.993	0.971–1.017	0.578	1.018	0.992–1.045	0.171
	T3-T4 vs. T1-T2	5.158	2.675–9.947	<0.001	7.658	3.280–17.876	<0.001
	N1 vs. N0	1.241	0.653–2.360	0.510	1.686	0.805–3.530	0.166
	G3 vs. G1-2	1.009	0.538–1.894	0.977	0.819	0.405–1.654	0.577
**Rs11536898**	C allele + vs. C -	0.291	0.086–0.987	**0.048**	0.274	0.078–0.959	**0.043**
	Age at diagnosis	0.994	0.971–1.017	0.612	1.018	0.992–1.044	0.176
	T3-T4 vs. T1-T2	5.077	2.645–9.747	0.000	7.694	3.298–17.951	0.000
	N1 vs. N0	1.232	0.648–2.342	0.525	1.694	0.810–3.540	0.161
	G3 vs. G1-2	1.018	0.543–1.907	0.955	0.817	0.405–1.651	0.574

## Data Availability

The data presented in this study are available on request from the corresponding author.

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
