# Peer review of "Pathomorphological Manifestations and the Course of the Cervical Cancer Disease Determined by Variations in the TLR4 Gene"

_diagnostics, 2023, doi:10.3390/diagnostics13121999_

Round 1

Reviewer 1 Report

The manuscript submitted by Žilienė and coauthors shows the “Pathomorphological Manifestations and the Course of the Cervical Cancer Disease Determined by Variations in the TLR4 gene”. The article is original, well structured; easy to read with main emphasis on the prevalence of SNPs identification of TLR4 gene in 172 cervical carcinoma patients. In my opinion, the manuscript can be published in this journal, after the authors have addressed the following minor issues:

·       Authors are suggested to improve the introduction section by shorten the content and focusing on the main significance of this research.

·       Authors are suggested to improve the results by simplify the content.

·       Authors are requested to shorten the discussion section and correlate the findings with previous studies.

·       At few places, spelling correction and grammatical proofreading is required.

·       Authors are suggested to improve the conclusion by adding future prospective as per the standard format.

·       The authors are suggested to dually check the citations throughout the manuscript.

Author Response

Response to Reviewer 1 Comments

General comment: The manuscript submitted by Žilienė and coauthors shows the “Pathomorphological Manifestations and the Course of the Cervical Cancer Disease Determined by Variations in the TLR4 gene”. The article is original, well structured; easy to read with main emphasis on the prevalence of SNPs identification of TLR4 gene in 172 cervical carcinoma patients. In my opinion, the manuscript can be published in this journal, after the authors have addressed the following minor issues:

  • Authors are suggested to improve the introduction section by shorten the content and focusing on the main significance of this research.
  • Authors are suggested to improve the results by simplify the content.
  • Authors are requested to shorten the discussion section and correlate the findings with previous studies.
  • At few places, spelling correction and grammatical proofreading is required.
  • Authors are suggested to improve the conclusion by adding future prospective as per the standard format.
  • The authors are suggested to dually check the citations throughout the manuscript.

Response: Thank you for acknowledging the strength of the study and for your very positive comments.

Point 1: Authors are suggested to improve the introduction section by shortening the content and focusing on the main significance of this research.

Response 1: We have shortened the introduction to be more concise and go straight to the point on the main significance.

Point 2: Authors are suggested to improve the results by simplifying the content.

Response 2: We appreciate this suggestion and corrected the results. Please notice that we have adjusted the results based on specific comments from the observation of another reviewer to make the results clearer.

Point 3: Authors are requested to shorten the discussion section and correlate the findings with previous studies.

Response 3: We have shortened the discussion section. However, no single publication meets the objectives and results of our study, so the review in the discussion is an analytical assessment of trends in selected polymorphisms.

Point 4: At few places, spelling correction and grammatical proofreading is required.

Response 4: Thank you for this observation; an English-speaking translator has rechecked the manuscript.

Point 5: Authors are suggested to improve the conclusion by adding future prospective as per the standard format.

Response 5: Thank you for this suggestion and corrected the conclusion by adding future prospective (lines 501-503).

Point 6: The authors are suggested to dually check the citations throughout the manuscript.

Response 6: We appreciate this suggestion and checked the citations throughout the manuscript.

Reviewer 2 Report

Review

1.       Include statistics of cervical cancer

2.       Give a short note about cervical cancer in the introduction part

3.       Explain elaborately, how did you select TLR4 gene for your study

4.       Include the limitations and future perspective of your study

5.       In introduction part, last paragraph should be clear with your objective

6.       In methodology part, give a separate sub-heading for the survival analysis

Author Response

Response to Reviewer 2 Comments

Response: Thank you for your review and positive feedback. Based on your suggestions, we have made a few changes, which you will find below our point-by-point answers.

Point 1: Include statistics on cervical cancer.

Response 1: we appreciate this suggestion; we have integrated some statistics in the introduction sections (lines 83-86).

Point 2: Give a short note about cervical cancer in the introduction part.

Response 2: In the introductory part, we briefly described cervical cancer. (lines 81-82).

Point 3: Explain elaborately, how did you select the TLR4 gene for your study.

Response 3: Thank you for this suggestion; in the last paragraph of the introduction we have explained why we chose the TLR4 gene. (lines 111-113).

Point 4: Include the limitations and future perspective of your study.

Response 4: We are aware of the limited power of our analysis, as is often revealed in retrospective studies due to limited sample sizes, and have acknowledged this issue in the discussion section. We have included the limitations and future perspective of this study. (lines 292-297).

Point 5:  In the introduction part, the last paragraph should be clear with your objective.

Response 5: We appreciate this suggestion, and we have corrected the aim of the study at the end of the introduction accordingly.

Point 6: In the methodology part, give a separate sub-heading for the survival analysis.

Response 6: To comply with the proposal, we consider it sufficient to separate the survival section in clause 3.3.

Reviewer 3 Report

This retrospective study evaluated the role of TLR4 SNPs on the prognosis of CC. It is an interesting paper. I would suggest the authors to shorten the introduction and discussion sections. In addition, the authors should further discuss the possible clinical implications of this study. The patient selection should be further explained as it is not clear whether consecutive patients were included? What were the exclusion criteria? I have no other comments.

Author Response

Response to Reviewer 3 Comments

General comment: This retrospective study evaluated the role of TLR4 SNPs on the prognosis of CC. It is an interesting paper. I would suggest the authors to shorten the introduction and discussion sections. In addition, the authors should further discuss the possible clinical implications of this study. The patient selection should be further explained as it is not clear whether consecutive patients were included? What were the exclusion criteria? I have no other comments.

Response: Thank you very much for your review and for acknowledging the strength of the paper. To your suggestion to shorten the introduction and discussion sections, we have shortened the introduction to be more concise and go straight to the point on the main significance, and we have shortened the discussion section. However, no single publication meets the objectives and results of our study, so the review in the discussion is an analytical assessment of trends in selected polymorphisms. We appreciate the suggestion to discuss the possible clinical implications of this study and have made some changes to the discussion section to reflect the potential benefit of the study (lines 289-291). We have added more information about patients‘ inclusion and exclusion criteria. Consecutive patients with stages I-IV cervical cancer who agreed to participate in the clinical hospital of the Lithuanian University of Health Sciences were included in the retrospective study. Patient exclusion criteria were other malignancies and incomplete medical documentation (lines 123-127).

Reviewer 4 Report

A very long manuscript with many references. It is not easy and friendly to read in the Results. Certain major concerns are in the manuscript. It is difficult to interpret the result that A allele of rs11536898 was significantly more frequent in patients with metastasis from the Tables.

Legend description of Table.4 is not enough, the abbreviations are not mentioned. What is M1 and M0?

Most of the content of Discussion addressed the SNPs with other cancers, authors should discuss more about the relationship of SNPs with disease progression and clinical status as the title stated.

Authors are advised to check the typing mistakes and grammatical errors in the text.

Author Response

Response to Reviewer 4 Comments

Response: Thank you very much for your review and for acknowledging the strength of the paper. Based on your suggestions, we have made a few changes, which you will find below our point-by-point answers.

Point 1:  A very long manuscript with many references. It is not easy and friendly to read in the Results. Certain major concerns are in the manuscript. It is difficult to interpret the result that the A allele of rs11536898 was significantly more frequent in patients with metastasis from the Tables.

Response 1: We agree with the reviewer about the long manuscript with many references. Many publications are analyzing the TLR4 gene and its links with cancer and other diseases. It was important for us to analyze all published information and present it in its entirety, which may be useful in summarizing the potential effects of selected polymorphisms on cancer. However, no single publication meets the objectives and results of our study, so the review in the discussion is an analytical assessment of trends in selected polymorphisms. We have shortened the introduction trying to be more concise, and we have shortened the discussion section.

Thank you for allowing us to clarify the results. We have modified the results accordingly, including explanations. We found a couple of technical mistakes and corrected them (lines 193-198, 253-263).

The allelic model suggested that the carriers of rs11536898 A allele had 5.068 times increased risk for distal metastases than the non-carriers. This was partially confirmed by the genotype model as patients with the CA genotype had a 4.735 higher risk for distal metastases than patients with the CC genotype. In the case of patients with the AA genotype vs. those with CC, no significant associations with metastases were determined. This may be due to the fact that only five patients with the AA genotype were determined in our study, which may have affected the p-value in this comparison.

It was determined that patients with the AA genotype were predisposed to shorter PFS than patients with CC. No significant effect of the CA genotype on PFS was determined. This is in line with the allelic model, which demonstrated that the carriers of the C allele were less likely to have shorted PFS when compared to the non-carriers.

Additionally, if we consider our results from the logistic regression analyses, the data suggest the deleterious role of AA and CA genotypes in metastases formation.

Point 2:  Legend description of Table.4 is not enough, the abbreviations are not mentioned. What is M1 and M0?

Response 2: we appreciate this suggestion, and have mentioned the abbreviations below the table.

Point 3:  Most of the content of the Discussion addressed the SNPs with other cancers, authors should discuss more about the relationship of SNPs with disease progression and clinical status as the title stated.

Response 3:  To follow the reviewer's suggestion, we have shortened the discussion section.

Point 4: Authors are advised to check the typing mistakes and grammatical errors in the text.

Response 4:  thank you for this observation; an English-speaking translator has rechecked the manuscript.

Round 2

Reviewer 4 Report

The manuscript has been improved. No further questions.